# Ray-Tracing-Based Numerical Assessment of the Spatiotemporal Duty Cycle of 5G Massive MIMO in an Outdoor Urban Environment

**Sergei Shikhantsov *,†** , **Arno Thielens, Sam Aerts** , **Leen Verloock, Guy Torfs, Luc Martens,** **Piet Demeester and Wout Joseph**

Department of Information Technology, Ghent University/IMEC, 9052 Ghent, Belgium;
arno.thielens@ugent.be (A.T.); sam.aerts@ugent.be (S.A.); leen.verloock@ugent.be (L.V.);
guy.torfs@ugent.be (G.T.); luc1.martens@ugent.be (L.M.); piet.demeester@ugent.be (P.D.);
wout.joseph@ugent.be (W.J.)
* Correspondence: sergei.shikhantsov@ugent.be
† Current address: Technologiepark-Zwijnaarde 126, B-9052 Ghent, Belgium.

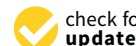

**Featured Application: The presented numerical approach can be directly applied to the estimation of the compliance boundary of the antenna array base stations and the downlink human EMF exposure assessment in the networks served by such base stations.**

**Abstract:** In the near future, wireless coverage will be provided by the base stations equipped with dynamically-controlled massive phased antenna arrays that direct the transmission towards the user. This contribution describes a computational method to estimate realistic maximum power levels produced by such base stations, in terms of the time-averaged normalized antenna array gain. The Ray-Tracing method is used to simulate the electromagnetic field (EMF) propagation in an urban outdoor macro-cell environment model. The model geometry entities are generated stochastically, which allowed generalization of the results through statistical analysis. Multiple modes of the base station operation are compared: from LTE multi-user codebook beamforming to the more advanced Maximum Ratio and Zero-Forcing precoding schemes foreseen to be implemented in the massive Multiple-Input Multiple-Output (MIMO) communication protocols. The influence of the antenna array size, from 4 up to 100 elements, in a square planar arrangement is studied. For a 64-element array, the 95th percentile of the maximum time-averaged array gain amounts to around 20% of the theoretical maximum, using the Maximum Ratio precoding with 5 simultaneously connected users, assuming a 10 s connection duration per user. Connection between the average array gain and actual EMF levels in the environment is drawn and its implications on the human exposure in the next generation networks are discussed.

**Keywords:** 5G; EMF exposure; Ray-Tracing; Massive MIMO

---

## 1. Introduction

The much anticipated roll-out of the fifth-generation (5G) telecommunication networks brings about new challenges associated with limiting the exposure of the general population and workers to electromagnetic fields (EMF). One of the key universal features of the next-generation networks, shared among various 5G technologies, is the use of large antenna arrays at the base station (BS) side [1]. The radiation pattern of an antenna array depends on the amplitude and phase ratios of the array elements. By selecting the elements' amplitudes and phases in a specific way, a BS can produce directed "beams" in its far-field—the main lobes of the array radiation pattern. This technique is

referred to as beamforming. Precoding is another transmission technique that dynamically sets the antenna element coefficients to fulfill a predefined optimization objective. The more elements the antenna array has (up to a certain limit), the more narrow the beams it is capable of forming. A narrower beam means a higher maximum gain for equal total transmit power, as the EMF energy gets focused more tightly in the desired direction. Several different beamforming and precoding techniques are discussed below.

Multi-User Multiple-Input Multiple-Output (MU-MIMO) with codebook-based beamforming simultaneously generates a subset of a predefined set of beams (the "codebook") at a time. A user equipment (UE) uplink (UL) signaling is processed at the BS to choose the beam that best reaches the UE location. In line-of-sight (LOS) conditions, this typically is the beam which has the closest direction-of-departure (DOD) to the true direction to the UE. However, in non-LOS (NLOS), the largest portion of the EMF power might reach the UE through interactions with the propagation environment (e.g., reflections and diffractions), and thus the DOD of the beam chosen by codebook-based precoding does not necessarily correlate with the path of the shortest distance to the UE.

The Massive MIMO technology utilizes full channel state information (CSI) to multiplex multiple UEs in the same time-frequency resource, that is, to simultaneously transmit multiple data streams through separate spatial channels at a shared frequency. Massive MIMO leverages the knowledge of a continuously updated channel to *precode* the transmitted signal.

The Maximum Ratio Transmission (MRT) precoding scheme aims at maximizing the signal to noise ratio (SNR) at the target UEs [2]. If the UE in focus has a dominant LOS path, this results in forming a precisely directed single beam. If the UE is in a deep shadow region with multiple scattering paths, the BS distributes the available power over all these paths. This results in a less directive array pattern that forms a compact region with an elevated EMF around the target UE, a so-called *hot-spot*. The Zero Forcing (ZF) on the other hand, aims to maximize the signal to interference ratio (SIR) at the target UEs. If the target UEs' channels correlate considerably, a large portion of the BS resources is spent for interference mitigation. By doing so, unintended hot-spots might be created in the regions of the environment without active UEs. The effect of these precoding schemes on the power distributions in the angular domain serviced by the BS will be investigated in the following sections.

Human EMF exposure in the far-field is directly related to the incident electric field (E-field) strength. The EMF values are typically averaged over time. The International Commission on Non-Ionizing Radiation (ICNIRP) specifies a time duration of 6 min to be used for the EMF averaging in compliance assessment [3,4]. The averaging interval of 30 min is specified for the whole-body surface area of the E-field averaging in the most recent version of the guidelines [3]. In free space the E-field can be directly derived from the antenna gain, which is used in practice to establish the BS compliance boundary [5]. Actual E-field values observed in an environment are influenced by propagation or blockage of the transmitted signals. Nevertheless, the time-averaged array gain is a meaningful indicator of the typical EMF exposure induced by a BS. As array antennas adapt their radiation patterns to the environment, it is essential to include this inherently dynamic attribute in the modeling. In this article, we compare codebook beamforming and the aforementioned Massive MIMO transmission precoding schemes in terms of the maximum time-averaged antenna array gain they yield to distill conclusions about their effect on the EMF exposure in the serviced area.

This problem was first addressed in [6] for a single antenna array performing codebook-based beamforming. Several analytically-defined DOD distributions, acting as a proxy for the distributions of users inside the serviced area (i.e., uniform, cosine in elevation, and azimuth) were analyzed. All UEs were modeled in free space, thus having the LOS to the BS (no environment model was included). An analytic network utilization model was implemented to determine the number of simultaneously active UEs. The results showed around 6 dB reduction for the 95th percentile of the maximum time-average BS gain as compared to the theoretical maximum, for high system utilization values.

In [7], the approach of the authors of [6] was extended to a BS capable of MRT precoding (dubbed eigen-beamforming [7] or conjugate beamforming [8]). The 3GPP statistical model [9] was used with an urban environment, a single 64-element BS, and both indoor and outdoor UEs. Depending on the number of simultaneously served UEs, the 95th percentile of the compliance distance (which is proportional to the maximum time-averaged BS gain [5]) constituted 30% to 50% of the theoretical maximum.

This contribution builds upon and extends the approach proposed in [6,7] to more realistic precoding schemes and a more advanced Ray-Tracing (RT) channel model. The RT modeling yields *spatially consistent* channels, which depends on the environment geometry and UEs locations. For the MU-MIMO systems one important implication is a realistic UE DOD distribution, which governs the beam directions. The magnitude of the inter-UE channel correlation depending on the distance between the UEs is captured in the RT modeling—a factor that greatly impacts the BS pattern when using interference-canceling precoding schemes such as ZF. It has been shown that the RT method reproduces key parameters of measured Massive MIMO channels [10], whereas the state-of-the-art statistical model (WINNER-II) tends to underestimate the amount of correlation in the channels of closely spaced UEs [11]. Various other approaches to 5G channel modeling have been proposed in the literature and we direct an interested reader to the recent overview articles in [12,13].

## 2. Materials and Methods

In this section, we describe the environment model used in the RT simulations and methods for the results processing. An overview of the beamforming and precoding schemes is given.

### 2.1. Environment Model

The RT model consists of the environment geometry description and the transmitter–receiver (Tx–Rx) parameters. A geometrical entity is represented by the coordinates of its boundary faces (polygons) and the dielectric parameters (relative permittivity $\varepsilon_r$ and conductivity $\sigma$) of each face. A Tx (Rx) antenna is defined with its location, radiation pattern, and the carrier frequency. For every Tx antenna, *rays* are launched from its location in directions (nearly) uniformly distributed on a surface of a sphere centered at the Tx [14]. A ray is an abstraction which represents a flat wavefront, described with its (complex) EMF amplitudes and the propagation direction. The rays are propagated (traced) through the environment and their interactions (reflections, diffractions and transmissions), as well as path loss (PL) and time-of-flight, are tracked by the ray-tracer. If a ray passes sufficiently close to an Rx location, it is considered to be received and its state is recorded. The output of the simulation is a set of received rays for every defined Tx-Rx pair.

The RT output is site-specific, i.e., the channel between a fixed Tx-Rx pair depends on the surrounding geometry. To generalize the results a number of geometrical entities was generated stochastically, based on a few macroscopic parameters. Each realization of the environment geometry we call an *environment sample* [15]. Figure 1 presents one of the environment samples obtained from the model we describe below.

We simulate an outdoor urban macrocell bounded by a fixed flat square area 100 m by 100 m in size. Building blocks are represented by cuboids, width and length of which are sampled from a uniform random distribution in range from 15 m to 25 m. The height of a building block is drawn from a uniform random distribution in range from 5 m to 20 m. The buildings are positioned on a rectilinear grid, such that any two neighboring blocks are separated by exactly one empty grid cell. Rows of building blocks and straight lanes form a Manhattan-like urban city landscape. The spacing (lane) width is set equal to 10 m, 15 m, or 20 m randomly with equal probability. The dielectric properties of the cuboids model concrete material with $\varepsilon_r = 7$, $\sigma = 1.5 \times 10^{-2}$ S/m. The ground plane is assigned asphalt dielectric properties with $\varepsilon_r = 5.7$, $\sigma = 5 \times 10^{-4}$ S/m.

The locations and properties of the Tx and Rx antennas are fixed in the model. The simulations are performed at a single frequency of 3.5 GHz, foreseen to be heavily used in 5G networks.

The BS (Tx) is a rectangular 10-by-10 element array of vertically polarized half-wave dipole antennas with a half-wavelength uniform inter-element spacing ($\lambda \simeq 85$ mm at 3.5 GHz). The center of the Tx array is positioned at $x = 1$ m, $y = 50$ m and $z = 25$ m (Figure 1). The BS height of 25 m correspond to the macrocell scenario in the 3GPP model [9].

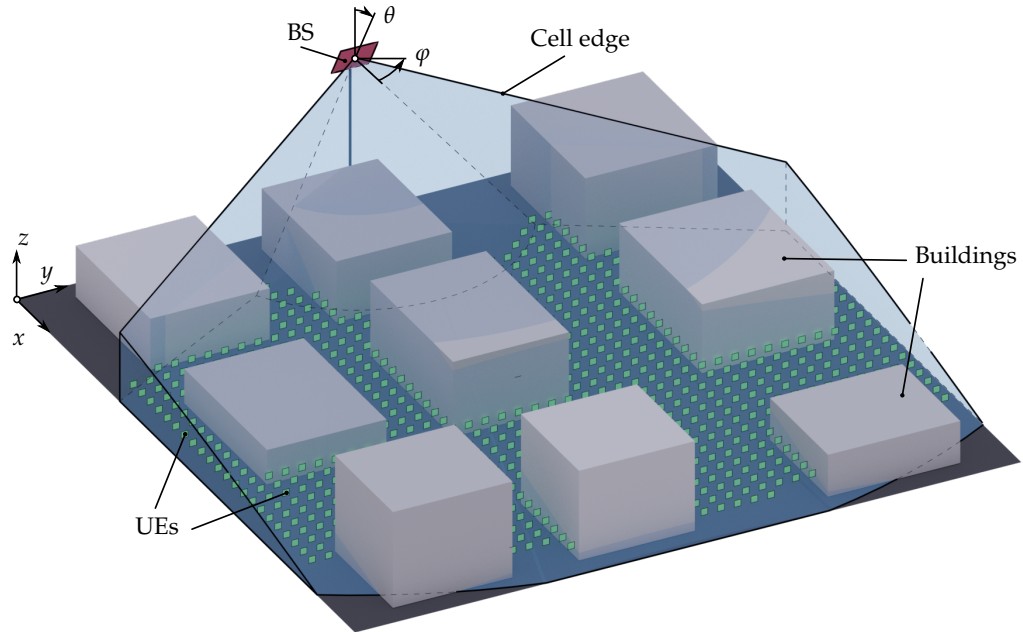

**Figure 1.** An example of an environment sample. The BS array is depicted in red. The cell boundaries are shown in blue. The Rx locations are shown in green.

In the following simulations, the BS coverage range spans from $-60°$ to $+60°$ in azimuthal ($\varphi$) and from $105°$ to $135°$ in polar angular directions ($\theta$) in coordinate system shown in Figure 1. This is in accordance to the model used in [6]. To include most of the simulated ground-plane area within its coverage range, the BS array is tilted down by $30°$ around the $y$-axis through its center.

A UE (Rx) is modeled as a single-terminal device equipped with a vertically-oriented vertically-polarized half-wave dipole antenna. The UEs are arranged on a regular rectilinear grid with 2 m spacing in $x$ and $y$ directions, at a height of 1.5 m above the ground-plane ($z$-axis). Only the grid nodes that fall within the cell and are located outside the building block interiors are kept in the simulation, which, on average, results in around 600 UEs per simulation. These UEs are used as potential active receivers in the analysis described in Section 2.3.

The density of the UE locations obtained in 25 environment samples is presented in Figure 2. The ($\varphi, \theta$) coordinate system is shown in Figure 1. Figure 2 describes the UE locations in the DOD space, as viewed from the BS. The UE distribution averaged over the environment samples is symmetric with respect to the plane $\varphi = 0$, as a result of the matching (statistical) symmetry of the environment geometry. The UE density increases towards the upper cell edge (further away from the BS) as the polar angle of incidence decreases.

The RT simulation parameters were set as recommended in [16], limiting the environment interactions to up to 6 reflections, 1 diffraction, and 1 transmission.

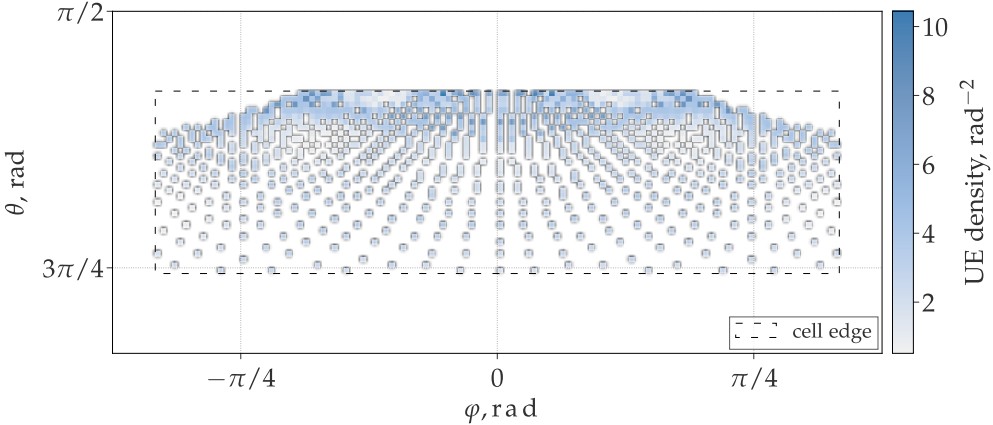

**Figure 2.** User equipment (UE) direction-of-departure (DOD) density averaged over 25 environment samples, viewed from the center of the BS array. The dashed line marks the cell limits.

### 2.2. MIMO Channel Matrix, Beamforming and Precoding

By $s_{nk}, n \in [1, 2, \ldots, N], k \in [1, 2, \ldots, K]$ we denote a set of rays between $n$th Tx and $k$th Rx points, with $N$ and $K$ being the total number of the Tx and Rx antennas, respectively. Then, the channel coefficient $\hat{h}_{kn}$ is given by (see, e.g., in [17])

$$\hat{h}_{kn} = \sum_{r \in s_{nk}} u_r \exp(-2\pi i f \tau_r), \tag{1}$$

where $u_r$ is the voltage amplitude induced by the ray $r$ at the Rx antenna terminal and $\tau_r$ is the time-of-flight of the ray $r$. The MIMO channel matrix $\hat{\mathbf{H}}$ is obtained by evaluating (1) for each Tx–Rx pair in the simulation. Columns $\hat{\mathbf{h}}_k$ of $\hat{\mathbf{H}}$ are channel vectors to the Rx with index $k$. The distance from different UEs to the BS may vary significantly. Therefore, UEs will experience differences in PL in comparison to one another. Thus, channel equalization is performed by normalizing $\hat{\mathbf{H}}$ column-wise [2]

$$\mathbf{H} = [\frac{\mathbf{h}_1}{||\mathbf{h}_1||}, \frac{\mathbf{h}_2}{||\mathbf{h}_2||}, \ldots, \frac{\mathbf{h}_K}{||\mathbf{h}_K||}], \tag{2}$$

where $|| \cdot ||$ denotes the Frobenius norm.

Having the normalized channel matrix, the Massive MIMO precoding matrices $\mathbf{W}$ are given by [8]

$$\mathbf{W} = \begin{cases} \alpha \mathbf{H}^H, & \text{for MRT,} \\ \alpha \mathbf{H}^H (\mathbf{H}\mathbf{H}^H)^{-1}, & \text{for ZF,} \end{cases} \tag{3}$$

where $(\cdot)^H$ denotes the Hermitian transpose and $\alpha$ is a real-valued normalization coefficient, chosen such that $\mathbf{W}$ has unit Frobenius norm.

Furthermore, the codebook steering matrix $\mathbf{W}_{CB}$ is constructed from the steering column-vectors $\mathbf{b}_k \in \mathbb{C}^N$ as [18]

$$\mathbf{b}_k = [\exp(2\pi i(\mathbf{d}_1, \mathbf{c}_k)/\lambda), \exp(2\pi i(\mathbf{d}_2, \mathbf{c}_k)/\lambda), \ldots, \exp(2\pi i(\mathbf{d}_N, \mathbf{c}_k)/\lambda)]^T, \tag{4}$$

$$\mathbf{W}_{CB} = \alpha[\mathbf{b}_1, \mathbf{b}_2, \ldots, \mathbf{b}_K]. \tag{5}$$

In (4), $\mathbf{d}_n$ is a distance vector from the BS array center to the $n$th element, $\mathbf{c}_k$ is a unit vector in the codebook direction assigned to the $k$th UE, and $(\cdot, \cdot)$ denotes the dot product. Knowing the channel vector $\hat{\mathbf{h}}_k$, the beamforming direction $\mathbf{c}_k$ is chosen as

$$\mathbf{c}_k = \underset{\{\mathbf{c}_i\}}{\arg \max} \, [(\hat{\mathbf{h}}_k, \mathbf{b}_i)]. \tag{6}$$

In (6), the maximization is carried out over the set of all beamforming directions $\mathbf{c}_i$ supported by the BS. The beamforming direction vector $\mathbf{c}_i$ is a unit vector in the direction $(\theta_i, \varphi_i)$ of the *i*th beam center

$$\mathbf{c}_i = \mathbf{c}(\theta_i, \varphi_i), \tag{7}$$

$$\mathbf{c}(\theta, \varphi) = [\sin\theta\sin\varphi, \sin\theta\sin\varphi, \cos\theta]^T, \tag{8}$$

where $\theta$ and $\varphi$ are the polar and azimuthal angles, respectively, in the spherical coordinate system depicted at Figure 1. The modeled system differentiates 32 beam directions in azimuth and 8 in elevation.

The transmit vector $\mathbf{t} \in \mathbb{C}^N$ is obtained by multiplying the precoding or steering matrix by the vector of transmitted symbols $\mathbf{s}$

$$\mathbf{t} = \mathbf{Ws}. \tag{9}$$

As the EMF is further assessed in terms of the time-average root mean square (RMS) values, with no loss of generality we set all transmitted symbols to be real-valued. In addition, we assume that no per-user power management is implemented at the BS and equal share of transmit power is directed to each UE. Therefore, we define $\mathbf{s} = [\sqrt{1/K}, \sqrt{1/K}, \dots, \sqrt{1/K}]^T$. The normalization $\sqrt{1/K}$ is needed for the transmit vector $\mathbf{t}$ in (9) with $\mathbf{W}$ given by (3) or (5) to satisfy the overall transmit power constrain.

### 2.3. Time-Average Antenna Array Patterns

An instantaneous array pattern is calculated as a sum of the patterns of its individual elements, weighted with the components of the transmit vector $\mathbf{t}$. As all antennas in the BS array are identical dipoles, this gives

$$A(\theta, \varphi, \mathbf{t}) = \sum_{n=1}^{N} A_{dip}(\theta, \varphi) t_n \exp(-2\pi i (\mathbf{d}_n, \mathbf{c}(\theta, \varphi)) / \lambda), \tag{10}$$

where $A_{dip}$ is a half-wave dipole radiation pattern [19], and $t_n$ denotes the *n*th element of $\mathbf{t}$. Here we do not account for the effect of mutual coupling in the antenna array, i.e., the modification of the free-space antenna element pattern by the currents in the neighboring elements.

In the far-field region of a BS, the incident EMF is proportional to the antenna gain in the direction where the measurement is preformed. As mentioned above, ICNIRP specifies [4] an EMF time-averaging interval $T_{avg} = 6$ min for the human exposure assessment. At the same time, it is foreseen that in a typical scenario 5G DL traffic will be transmitted in short bursts (in the order of tens of seconds [6]), switching between sets of UEs that demand it at any given moment. If the served UEs are distributed uniformly enough within the cell, the BS would focus the transmission in many different directions over a sufficiently long time interval. Therefore, a *realistic* time-average BS antenna array gain is expected to differ significantly from the theoretical maximum one.

To quantify how much the time-averaged gain is reduced relative to the theoretical maximum we follow the approach proposed in [6,7]. We introduce a constant *T*—the duration of one connection ("drop duration" in [7] or "scheduling time" in [6]). We model a network in which independent sets of *K* UEs are served for time *T* in series with no overlaps. Then, the time-average BS array radiation pattern $\tilde{A}_m^{N,K}(\theta, \varphi)$ is calculated as a weighted mean of the instantaneous patterns *i* produced during the averaging interval

$$\tilde{A}_m^{N,K}(\theta, \varphi) = \sum_i \omega_i A_m^{N,K}(\theta, \varphi, \mathbf{t}^i), \tag{11}$$

where $m \in \{CB, MRT, ZF\}$ denotes the transmission precoding scheme used at the BS; $N, K$ are the number of utilized antenna elements and the number of simultaneously served UEs, respectively, and $\omega_i$ is the fraction of the averaging time during which pattern $i$ was active, varying from 0 (not in the averaging interval) to $T/T_{avg}$ (fully inside the averaging interval). In the following, for convenience we choose $T$ such that $T_{avg}$ is its integer multiple, then $\omega_i = T/T_{avg}$. Next, the normalized gain $G_m^{N,K}$ is given by

$$G_m^{N,K}(T) = \frac{\max_{\theta,\varphi} [\tilde{A}_m^{N,K}(\theta, \varphi)]}{G_{max}^N}, \tag{12}$$

where $G_{max}^N$ is the maximum gain of an array of $N$ elements. The maximum gain of a planar antenna array is calculated as a product of the maximum antenna element gain and the maximum array factor. The maximum array factor equals to the number of elements in the array [20]. Therefore, for an antenna array composed of identical half-wave dipoles, $G_{max}^N = \max_{\theta,\varphi}[A_{dip}] \cdot N \simeq 1.64 \cdot N$. This value is further used in (12) as a normalization factor.

The RMS E-field strength at the location of the UE, to which $G_m^{N,K}$ would be directed, can be estimated using a free-space approximation according to the following expression,

$$E_{RMS} = \frac{\cos(\pi - \theta_{max})}{h_{BS} - h_{UE}} \sqrt{\frac{ZPG_m^{N,K}G_{max}^N}{2\pi}}, \tag{13}$$

where $\theta_{max}$ is the polar angle of $G_m^{N,K}$, $Z \simeq 377$ Ohm is the impedance of free space, $P$ is the BS total radiated power, and $h_{BS} = 25$ m and $h_{UE} = 1.5$ m are the BS and the UE height above the ground, respectively. For example, taking a BS with 64 antenna elements and 200 W nominal power [21], the highest achievable $E_{RMS}$ ($G_m^{N,K} = 1$) in the direction normal to the BS array plane ($\theta_{max} = 120°$) equals around 18.6 V/m, which falls below the ICNIRP reference level of 61 V/m [4].

To calculate $G_m^{N,K}(T)$ for given $m$, $T$, $N$, and $K$, a numerical experiment is performed. Figure 3 shows a flowchart describing the procedure.

We studied configurations with 2-by-2, 4-by-4, 6-by-6, and 8-by-8 square sub-arrays selected from the center of the simulated 10-by-10 Tx array, as well as the complete array itself. These correspond to total array counts $N$ of 4, 16, 36, 64, and 100 elements. Scenarios with $K = 1, 2, 5$, and 10 simultaneously active UEs were studied for each $N$. Connection duration $T_{avg}$ equal to 60 s, 10 s, and 1 s was considered for each $N$ and $K$. In total, $N_{env} = 25$ environment samples were simulated. Gain values obtained from the 25th sample were found to change the 95th percentiles of the time-averaged gain distributions by less than 1% (see Figure 5 and Section 4.2), which was accepted as a sufficient level of accuracy. Finally, in every environment sample $N_s = 100$ time-averaged gain evaluations were performed, which amounts to 2500 evaluations of $G_m^{N,K}(T)$ for each value of $N, K, T$ and precoding scheme $m$. In the following section we present and discuss the distributions of $G_m^{N,K}$.

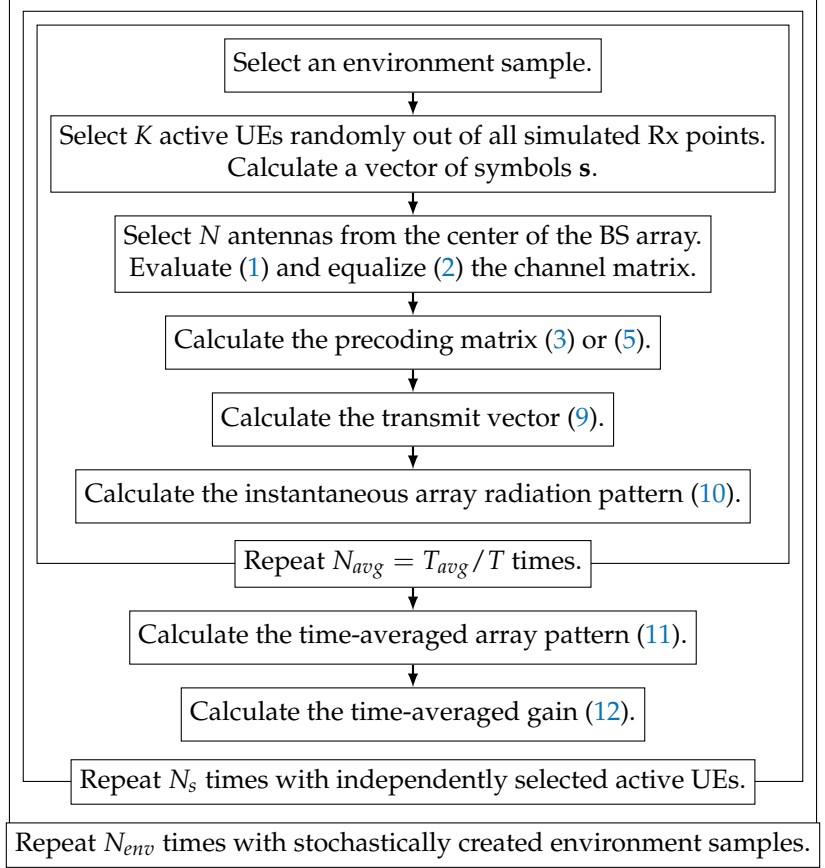

**Figure 3.** A flowchart of the procedure used to generate distributions of the time-averaged gain values $G_m^{N,K}(T)$. The complete procedure yields $N_{env} \cdot N_s$ time-averaged gain evaluations.

## 3. Results

### 3.1. Average Array Patterns

Figure 4 shows the DODs of $G_{CB}^{N,5}$ (fist column), $G_{MRT}^{N,5}$ (second column), and $G_{ZF}^{N,5}$ (third column), for $N = 4$ (first row), $N = 36$ (second row), and $N = 100$ (third row), calculated with a connection time $T = 60$ s. The DOD of each time-averaged gain sample is depicted with a black circle in the $(\varphi, \theta)$ coordinate system. The circle size is proportional to $G_m^{N,K}$, and its opacity is proportional to the number of samples observed at the corresponding DOD. The background pseudocolor plot shows the sample-average of $\tilde{A}_m^{N,K}(\theta, \varphi)$, illustrating the difference in the average beamwidths of the obtained patterns. $G_{ZF}^{4,5}$ is undefined, as the condition $N > K$ must be satisfied for $\mathbf{HH}^H$ to be invertible, which is necessary to calculate $G_{ZF}$ according to (3). Its plot was therefore not included in Figure 4.

### 3.2. Normalized Gain

Figure 5 shows the cumulative distribution functions (CDFs) of $G_m^{N,K}$ in the layout matching that of Figure 4. Each plot in Figure 5 presents a CDF for $K = 1$ (black), $K = 2$ (red), $K = 5$ (blue), and $K = 10$ (green). In addition, the 95th precentile of each CDF is marked with a vertical dashed line of the same color. Table 1 lists the 95th percentiles for all possible combinations of the studied parameters. Additionally, the table cell background color saturation is proportional to its numerical value, ranging from white for zero to deep blue for one. This gives a visual cue to how different parameter combinations affect the normalized time-averaged gain.

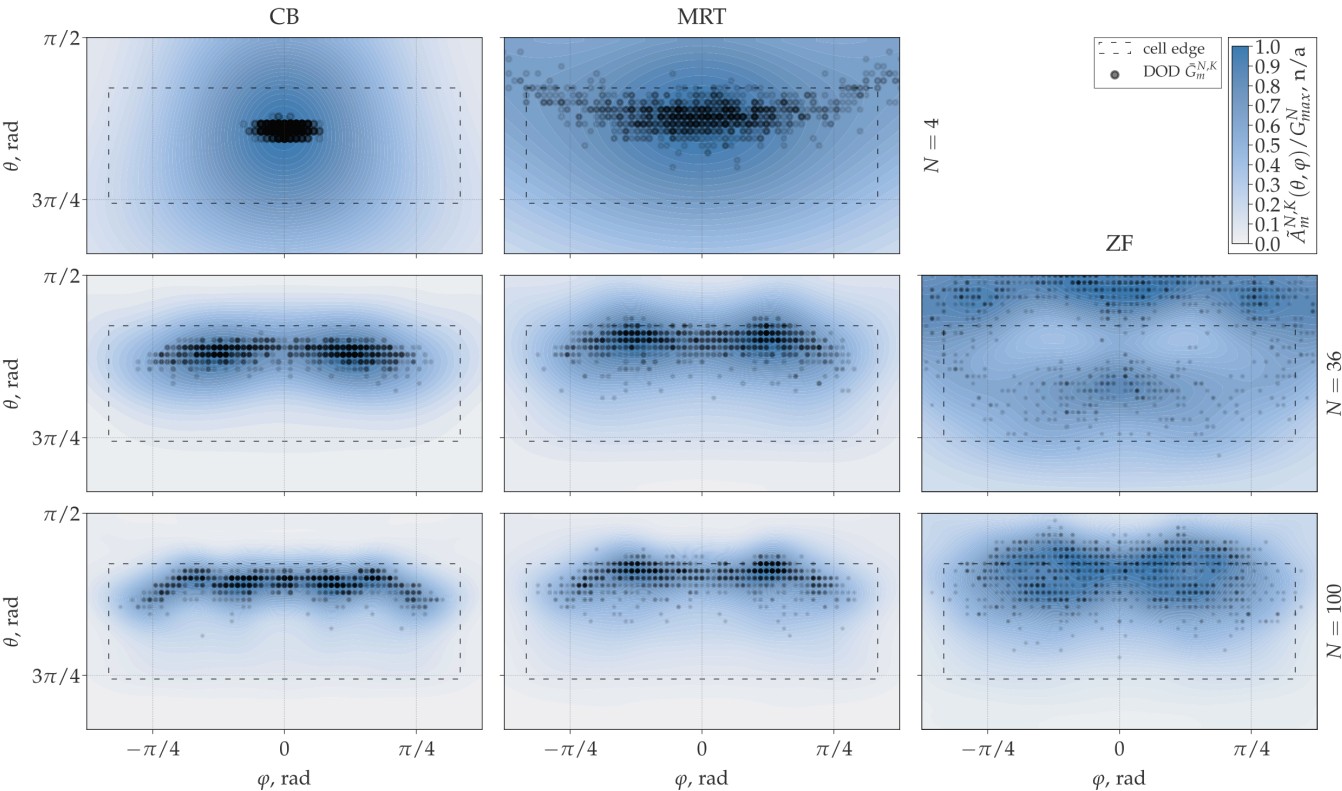

**Figure 4.** DOD $(\varphi, \theta)$ of the maxima of the time-averaged BS array patterns as observed over 2500 cases (100 simulations with randomly distributed UEs in 25 different environment samples) when serving $K = 5$ UEs simultanously, with connection time $T = 60$ s. Each maximum direction is marked with a black circle. The circle size is proportional to the corresponding value of the time-averaged maximum gain (normalized to the maximum of all samples of the respective parameter combination). The circle opacity is proportional to the number of maxima found in the corresponding $(\varphi, \theta)$ direction. Left, center, and right columns show data for CB, MRT, and ZF transmission schemes, respectively. In the first, second, and third rows, scenarios with 2-by-2, 6-by-6, and 10-by-10 base station arrays are depicted, respectively. The ZF transmission with 2-by-2 BS array ($N = 4$) is undefined and was omitted. The dashed line depicts the cell boundary. The normalized time- and sample-averaged BS array patterns are shown in blue.

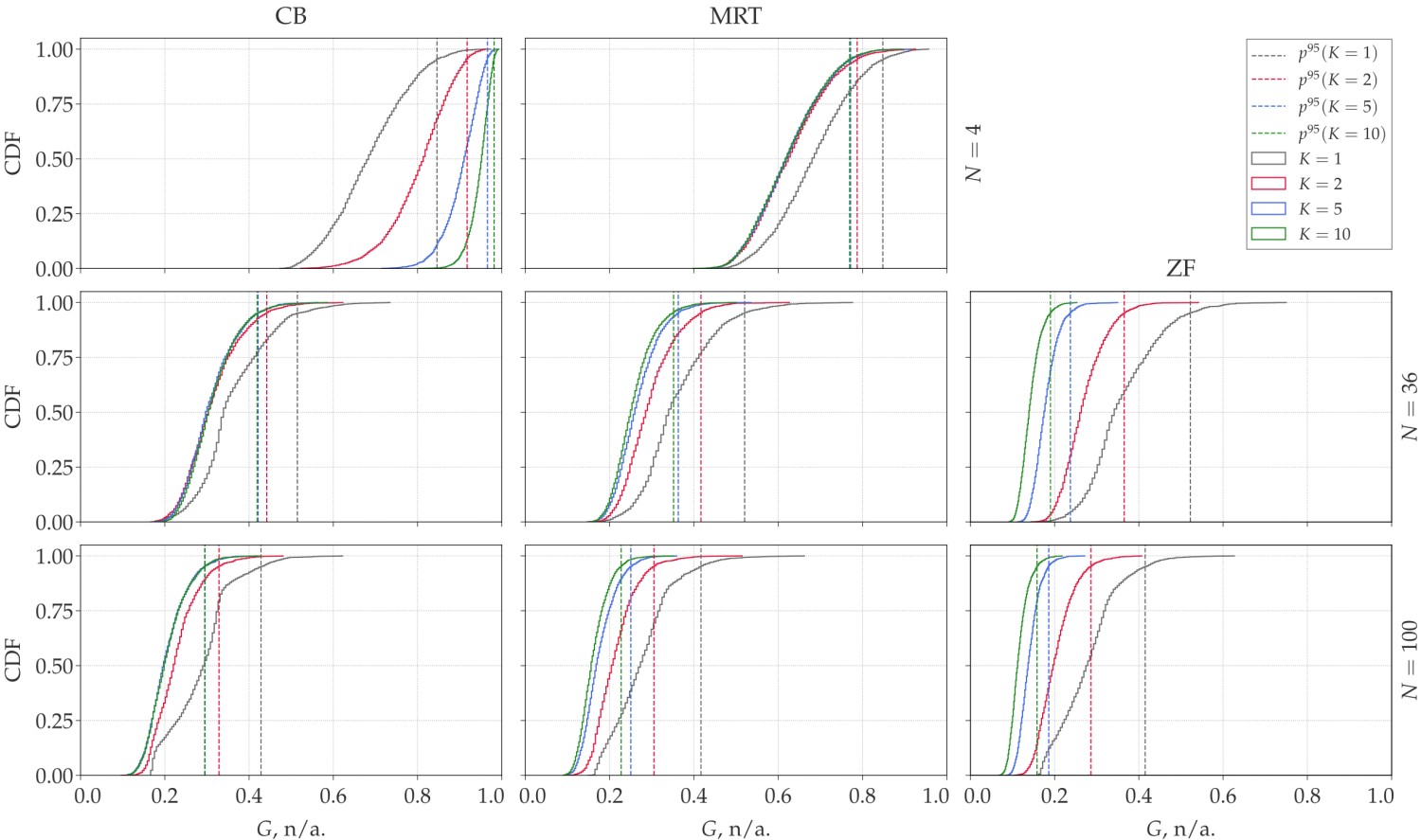

**Figure 5.** Cumulative distribution functions (CDFs) of the normalized 6 min average BS array gain $G_{CB}^{N,K}$ (fist column), $G_{MRT}^{N,K}$ (second column), $G_{ZF}^{N,K}$ (third column) with $T = 60$ s. Scenario with base station array size $N = 4$, 36, and 10 are shown in the first, second, and third rows, respectively. Solid lines show CDFs for the number of simultaneously active UEs $K = 1$ (black), $K = 2$ (red), $K = 5$ (blue), and $K = 100$ (green). Dashed lines of matching color mark the 95th percentile of each distribution.

**Table 1.** Summary of the 95th percentiles of $G_{CB}, G_{MRT}$, and $G_{ZF}$ for $T \in \{60\,\text{s}, 10\,\text{s}, 1\,\text{s}\}$, $K \in \{1, 2, 5, 10\}$, and $N \in \{4, 16, 36, 64, 100\}$. The background color saturation is proportional to its numerical value, ranging from white for zero to deep blue for one.

| Scheme | | CB | | | | | MRT | | | | | ZF | | | |
|---|---|---|---|---|---|---|---|---|---|---|---|---|---|---|---|
| K \ N | | 4 | 16 | 36 | 64 | 100 | 4 | 16 | 36 | 64 | 100 | 16 | 36 | 64 | 100 |
| $T = 60$ s | 1 | 0.85 | 0.63 | 0.51 | 0.46 | 0.43 | 0.85 | 0.63 | 0.52 | 0.47 | 0.42 | 0.63 | 0.52 | 0.46 | 0.42 |
| | 2 | 0.92 | 0.58 | 0.44 | 0.37 | 0.33 | 0.79 | 0.53 | 0.42 | 0.35 | 0.31 | 0.44 | 0.37 | 0.32 | 0.29 |
| | 5 | 0.97 | 0.57 | 0.42 | 0.35 | 0.29 | 0.77 | 0.49 | 0.36 | 0.30 | 0.25 | 0.29 | 0.24 | 0.21 | 0.19 |
| | 10 | 0.98 | 0.55 | 0.42 | 0.34 | 0.30 | 0.77 | 0.48 | 0.35 | 0.27 | 0.23 | 0.27 | 0.19 | 0.17 | 0.16 |
| $T = 10$ s | 1 | 0.70 | 0.40 | 0.29 | 0.24 | 0.20 | 0.71 | 0.42 | 0.31 | 0.25 | 0.22 | 0.42 | 0.31 | 0.26 | 0.21 |
| | 2 | 0.83 | 0.40 | 0.27 | 0.21 | 0.17 | 0.66 | 0.38 | 0.27 | 0.22 | 0.19 | 0.31 | 0.24 | 0.19 | 0.17 |
| | 5 | 0.92 | 0.42 | 0.28 | 0.22 | 0.18 | 0.65 | 0.36 | 0.27 | 0.20 | 0.16 | 0.20 | 0.15 | 0.13 | 0.11 |
| | 10 | 0.96 | 0.44 | 0.31 | 0.24 | 0.20 | 0.64 | 0.36 | 0.25 | 0.19 | 0.16 | 0.18 | 0.11 | 0.10 | 0.09 |
| $T = 1$ s | 1 | 0.66 | 0.32 | 0.21 | 0.17 | 0.13 | 0.67 | 0.36 | 0.26 | 0.20 | 0.15 | 0.35 | 0.25 | 0.19 | 0.16 |
| | 2 | 0.80 | 0.32 | 0.21 | 0.16 | 0.13 | 0.60 | 0.34 | 0.24 | 0.19 | 0.15 | 0.26 | 0.20 | 0.16 | 0.14 |
| | 5 | 0.90 | 0.37 | 0.25 | 0.18 | 0.14 | 0.60 | 0.33 | 0.23 | 0.18 | 0.14 | 0.16 | 0.12 | 0.10 | 0.09 |
| | 10 | 0.95 | 0.42 | 0.29 | 0.21 | 0.18 | 0.59 | 0.32 | 0.23 | 0.18 | 0.14 | 0.14 | 0.09 | 0.08 | 0.06 |

## 4. Discussion

### 4.1. Array Patterns for CB, MRT and ZF

The left column of Figure 4 shows the DOD of the time-averaged gain observed when applying codebook precoding. The maximum value of the time-averaged array pattern was always found within the cell boundary. This was expected, as any instantaneous single-user codebook pattern has its main lobe pointing approximately towards an active UE, which was always situated within the cell. As a result of the linearity of (5), (9), and (10) with respect to the steering vector **b**, the 6 min time-averaged pattern is expressed as an average of the instantaneous patterns towards the UEs served during the averaging interval. The maximum of such averaged pattern was most likely to be found at the intersection of the instantaneous array patterns, i.e., somewhere within the cell. In the scenario with a 2-by-2 BS array (top-left), the maximum tends to be located around the cell center in azimuth. The reason for that is the low directivity of a typical pattern produced by an array of only 4 elements. For $N = 36$ (center-left), then the maxima distribution follows the average UE density peaks (Figure 2), with two clusters that correspond to the lanes between the building blocks, parallel to the $x$-axis (Figure 1). With 100 BS antenna elements (bottom-left), finally, the gain maxima closely follow the regions of high UE density (Figure 2), nearly covering the full azimuth range of $120°$.

The center column of Figure 4 shows the DODs of the time-averaged gain found using MRT precoding. Similarly to CB, when the antenna count is low ($N = 4$, top-center plot), the maxima tend to be concentrated around the cell center. As $N$ increases, $G_{MRT}$ tends to be directed towards the regions densely occupied with the UEs with higher probability. However, MRT shows an increased spread of the gain DOD compared to CB. This can be attributed to the fact that unlike CB, which assigns a single beam per active UE, MRT superimposes a set of multiple beams with powers proportional to the contributions of the corresponding propagation paths to the total signal received by the UE. If a UE has a direct propagation path to the BS (i.e., LOS), the instantaneous MRT pattern is likely to have a global maximum in that direction (second strongest path—the ground reflection, if present, being orders of magnitude weaker). In case the target UE resides in a shadow region (NLOS), several propagation paths typically contribute comparable amounts to the total received signal, e.g., if a UE is

obstructed by a building, the main propagation mechanisms that make the connection possible are over-the-rooftop diffraction and reflections from the walls of the surrounding buildings. As a result, the time-average pattern maximum is sometimes found *outside* the cell boundary, as can be seen on the plots showing $G_{MRT}^{36,5}$ (center-center) and $G_{MRT}^{100,5}$ (bottom-center) in Figure 4.

This effect is even more pronounced when ZF precoding was used, although the underlying reason is different. ZF minimizes interference between the target UEs by canceling the transmission via shared paths. In scenarios with a large number of spatially correlated UEs, a portion of the total transmit power dedicated to fighting interference may exceed that of the intended signal, i.e., an instantaneous ZF pattern can have higher gain in its side-lobes than in the beams intended to reach the UEs. Such effect is observed in the DOD distribution of $G_{ZF}^{36,5}$ (center-right) in Figure 4. In the areas with the highest UE density, where both CB and MRT generally produced their time-average gain maxima most often, ZF showed very few time-average array pattern peaks. The ZF precoding efficiency generally increases with the $N/K$ ratio [2]. For $N = 100$ the gain distribution (bottom-right) was similar in shape to what was obtained using MRT, although the spread of the gain locations noticeably exceeded both MRT and CB.

*4.2. Normalized Time-Averaged Gain*

The CDFs in Figure 5 compare the $G_m^{N,K}$ values for parameter configurations that correspond to those shown in Figure 4. Increasing the BS antenna count $N$ decreases the normalized time-averaged gain for all studied schemes, with other parameters fixed. Two factors are contributing to this effect. First, the normalization coefficient in (12) is proportional to $N$, which counteracts the increase in the absolute array gain. Second, with larger $N$ the BS is capable of producing narrower beams, which are less likely to interlap in the DOD region within the cell, reducing the maximum of the average taken according to (11).

Decreasing the connection time $T$ also decreases the time averaged gain, that is, $G_m^{N,K}(T = 60\,\mathrm{s}) > G_m^{N,K}(T = 10\,\mathrm{s}) > G_m^{N,K}(T = 1\,\mathrm{s})$ for any fixed $m$, $N$ or $K$. This observation is explained by the fact that the more independent UE sets are served in the averaging time-span $T_{avg}$ (or, equivalently, the less the $T$ value is), the closer $G_m^{N,K}$ approaches the normalized average of instantaneous array patterns $A(\theta, \varphi, \mathbf{t})$ over the cell. Conversely, in the limit of a single UE served with $T = T_{avg}$, as follows from (10), (11), (12), the time-averaged gain is the instantaneous BS pattern maximum. In this case, the CB beamforming realizes the maximum theoretical gain $G_{max}$ for the codebook directions coinciding with the maxima of the BS antenna element's individual pattern. This can be seen by substituting $\mathbf{t} = \mathbf{b}_k$ into (10), assuming that the beam center DOD satisfies $(\theta_k, \varphi_k) = \arg\max_{\theta,\varphi}[A_{dip}]$.

When only a single UE is connected at a time ($K = 1$, shown in black in Figure 5), CB, MRT, and ZF show very similar distributions of the normalized gain. In fact, as can be seen from (3) in the degenerate case of $K = 1$, the matrix inverse of $\mathbf{H}\mathbf{H}^H$ becomes a reciprocal of a squared channel coefficient magnitude. The ZF and MRT formulations are then equivalent, with appropriately chosen normalization coefficients $\alpha$ in (3). The minor discrepancy in Table 1 between the MRT and ZF ($<1\%$) for $K = 1$ is due to the numerical round-off error propagation. The difference between $G_{MRT}^{N,1}$ and $G_{CB}^{N,1}$, gradually increases with increasing $N$ and decreasing $T$. Both $G_{MRT}^{4,1}$ and $G_{CB}^{4,1}$ are decreasing monotonously from around 0.85 for $T = 60\,\mathrm{s}$ to around 0.66 for for $T = 1\,\mathrm{s}$ (see Table 1). At $N = 100$, $G_m^{N,1}$ drops rapidly to around a half of that ($\simeq 0.42$ for CB) for $T = 60\,\mathrm{s}$, around a third ($\simeq 0.22$ for MRT) for $T \simeq 10\,\mathrm{s}$, and less than a quarter ($\simeq 0.15$ for MRT) for $T = 1\,\mathrm{s}$.

Increasing the number of simultaneously served UEs $K$ was found to decrease $G_{MRT}^{N,K}$ and $G_{ZF}^{N,K}$ for any fixed $N$ and $T$. Increasing $K$ decreased $G_{CB}^{N,K}$ only for larger BS arrays ($N \geq 16$) for $T = 60\,\mathrm{s}$ and led to its increase with any $N$ for shorter connection time values. For $N = 4$, the CB time-averaged gain closely approached the theoretical maximum for any $T$ and $K \geq 5$ ($G_{CB}^{N,K} \geq 0.9$). This indicates that smaller BS antenna arrays implementing CB beamforming offer little to no benefit in terms of human EMF exposure reduction.

In a realistic usage scenario, we take $T \simeq 10$ s [6]. If the BS is equipped with 64 antenna elements, $p^{95}$ of $G_{CB}^{64,K}$ is just above 0.2 (around 7 dB reduction) for any $K$. This is in agreement with the results in [7] obtained with a similar configuration in the outdoor macrocell environment. Direct comparison with [6] is not possible, as in that case the UE count was varied during the averaging time. Increasing the averaging time $T$ to 60 s increases the 95th percentile of $G_m^{64,5}$ to 0.35 (4.6 dB) for CB, 0.30 (5.2 dB) for MRT, and 0.21 (6.8 dB) for ZF.

Adding more BS antenna elements while using either the CB or MRT scheme, does not decrease $G_m^{N,K}$ significantly. Their lowest 95th percentile values were observed for $T = 1$ s: $G_{MRT}^{100,5} \simeq G_{MRT}^{100,10} \simeq 0.14$ and $G_{CB}^{100,1} \simeq G_{CB}^{100,2} \simeq 0.13$. The 95th percentiles of $G_{ZF}^{N,K}$ were lower than the respective $G_{MRT}^{N,K}$ and $G_{CB}^{N,K}$ values for all $K \geq 2$. This difference was larger with for larger $K$ and shorter connection time $T$. In the realistic scenario with $T = 10$ s, the 95th percentiles of $G_{ZF}^{64,K}$ were equal to around 0.19 (7.2 dB), 0.13 (8.9 dB), and 0.1 (10 dB) for $K = 2$, 5, and 10, respectively. These values are nearly two times lower that the MRT and CB schemes demonstrated in the same parameter configurations. The lowest $p^{95}$ was found with $G_{ZF}^{100,10} \simeq 0.06$ (12.2 dB), which is around one third of the corresponding CB value, and more than two times lower than the minimum for the MRT or CB scheme.

As the time-averaged gain is directly related the average $E_{RMS}$ measured at some location in the cell. The far-field instantaneous E-field magnitude is proportional to the square root of the antenna gain. Therefore, the time-averaged $E_{RMS}$ is reduced at least in proportion to the square root of the time-averaged gain $G_m^{N,K}$, relative to the $E_{RMS}$ estimate based on the maximum achievable gain $G_{max}^N$. In a scenario with $N = 64$, $K = 5$, and $T = 10$ s, this leads to the E-field reduction in 95% of the observations by at least a factor of around 2.1 (3.2 dB), 2.2 (3.4 dB), and 2.8 (4.5 dB) for the CB, MRT, and ZF schemes, respectively, compared to the theoretical maximum. The theoretical maximum gain value was never reached in samples with $K \geq 5$.

## 5. Conclusions

We presented a numerical approach that utilizes the RT method to model a time-averaged array gain of a 5G BS operating in a macrocell outdoor urban environment. The RT approach provides a more realistic signal propagation and user spatial correlation properties compared to analytical and stochastic approaches. In a realistic scenario, with a BS consisting of 64 antenna elements that serves 5 UEs simultaneously and a 10 s per user connection duration, 95% of the 6-min time-average gain observations fell below 0.22 (more than 6.6 dB reduction), 0.20 (7 dB) and 0.13 (8.9 dB) of the theoretical maximum, using codebook, maximum ratio transmission, and zero-forcing schemes, respectively. With user connection duration of 60 s, the corresponding 95th percentiles increase to 0.35 (4.6 dB), 0.30 (5.2 dB), and 0.21 (6.8 dB), respectively. In all studied scenarios, increasing the BS element count decreased the normalized time-average gain. With the MRT and ZF transmission schemes, lower time-averaged gain was always observed when the number of multiplexed UEs was increased. With the CB beamforming that was the case only for larger BS arrays. In all multi-user scenarios, the ZF yielded the lowest $p^{95}$ values of the normalized time-average gain (0.06 or 12.2 dB reduction with 100 BS antennas and 10 UEs), which is more than two times lower than any other studied precoding scheme.

**Author Contributions:** Conceptualization, S.S. and W.J.; methodology, L.V. and G.T.; software, S.S.; formal analysis, S.S.; resources, L.V. and S.A.; writing—original draft preparation, S.S.; writing—review and editing, A.T., S.A., and W.J.; visualization, S.S.; supervision, W.J., L.M., and P.D.; project administration, W.J., L.M., and P.D.; funding acquisition, W.J., L.M., and P.D. All authors have read and agreed to the published version of the manuscript.

**Funding:** Piet Demeester thanks the ERC for his advanced grant 695495 "ATTO: A new concept for ultra-high capacity wireless networks". Arno Thielens is an FWO senior postdoctoral fellow. Sam Aerts is an FWO postdoctoral fellow.

**Conflicts of Interest:** The authors declare no conflict of interest.

## Abbreviations

The following abbreviations are used in this manuscript.

| | |
|---|---|
| LTE | Long-Term Evolution |
| MIMO | Multiple-Input Multiple-Output |
| EMF | electromagnetic field |
| MRT | Maximum Ratio Transmission |
| ZF | Zero Forcing |
| CB | Codebook Beamforming |
| BS | base station |
| UE | user equipment |
| DOD | direction of departure |
| ICNIRP | The International Commission on Non-Ionizing Radiation |
| PL | path loss |
| RT | Ray-Tracing |
| Rx | receiver |
| Tx | transmitter |
| DL | downlink |

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
