# Peer review of "Ray-Tracing-Based Numerical Assessment of the Spatiotemporal Duty Cycle of 5G Massive MIMO in an Outdoor Urban Environment"

_applsci, doi:10.3390/app10217631_

Round 1
Reviewer 1 Report
The calculation method here-proposed provides a closer reality estimate of the maximum levels produced by these antenna arrays. Ray-Tracing method, by using stochastic generation, allows the generalization of the acquired results.
Reviewer 2 Report
Authors proposed assessment method of time-averaged EMF levels in the urban environment using ray-tracing method. Three modes of the base station operations, several antenna arrays, and connection duration were considered as variation in the exposure scenario. The time averaged again, i.e., the effective gain during the averaging time maximized in the user equipment, was used as metric. The results showed that the proposed approach may derived much lower value of the time averaged gain than previous works. The approach is technically sound. While, exposure EMF is some claims were not clearly explained. Could you please consider to following terms.
Please make clear what does “theoretical” maximum gain imply. Generally, an antenna gain cannot derived theoretically. The definition should be clarified.
The approach proposed in this paper is used to estimate actual exposure level. The exposure level depends on transmitted-radiation power and separation distance between the base station and user equipment. Please comment on how the exposure level (Electric field strength) can be derived using the proposed metric, i.e., time-averaged gain. The recommendation is using equation with the brief explanation.
Line 57–59; ICNIRP 2020 guideline had been cited here, and the time average constant is 30 minutes in this guideline. While, the assessment was conducted using 6 minutes in this paper.
Also, ICNIRP 1998 should be cited and separated with ICNIRP 2020 guideline. In addition, ICNIRP 1998 should be cited in Line 185.
Line 138; Form figure 2, the plot seems symmetric, but the symmetricity of the UE density is difficult to recognize by color plot. Further explanation is needed. In addition, what does “symmetric on average” imply?
Line 142–144; Authors commented that the number of environment samples were sufficient for the statistical convergence of the result, and commented to refer 3.2. But, there is no description about the statistical convergence. Please add the comments at 3.2.
Line 240–241; Authors showed CDF of the normalized gain defined by Eq.(12). Please make clear the parameters variated to derive the CDF.
Line 247–349; authors claimed that their approach was more realistic than the analytical and stochastic approaches. Pleas discuss why the proposed approach was superior than the stochastic approaches in the section of discussion by citing relevant references.
(minor comments or recommendations )
Line 113 “in [15, 25] m”; The other blankets should be used for the convenience to the reader.
Line 207–219; It is recommended to use flowchart to explain the assessment procedure.
Figure 3 and 4; what does “G, n/a” imply?
Reviewer 3 Report
The paper topic is good and timely. However, the literature review part is not comprehensive. It must be improved by adding relevant studies and researches with appropriate references. Here, I provide a couple of them and please find others:
1) Statistical Characterization of 3D Propagation Model for V2V Channels in Rectangular Tunnels (IEEE Antennas and Wireless Propagation Letters, 2017)
2) Channel Modeling and System Concepts for Future Terahertz Communications: Getting Ready for Advances Beyond 5G (IEEE Vehicular Technology Magazine, 2020)
And please provide some insights into how would you compare the differences between ray-tracing-based massive MIMO characterization compared to the direct channel measurement approach.
What is the benchmark of your comparison?
Round 2
Reviewer 2 Report
One comment regarding to the reply to the following;
-------------------------
Q: Figure 3 and 4; what does “G, n/a” imply?
A: n/a is an abbreviation for ’not applicable’. It is there to avoid confusing the dimensionless normalized gain with gains expressed in dB.
-------------------------
This imply that authors aim to show that "the unit of the measure is not available".
Expressing the unit after "," is not general and confuse the reader.
In addition, there is no explanation about the unit.
Thus, ",n/a" recommended to delete through the figures.